# Animal Welfare Considerations When Conducting OECD Test Guideline Inhalation and Toxicokinetic Studies for Nanomaterials

**DOI:** 10.3390/ani12233305

**Published:** 2022-11-26

**Authors:** Yong Hyun Chung, Mary Gulumian, Richard C. Pleus, Il Je Yu

**Affiliations:** 1H&H Bio, Hoseo-ro 79-20, Baebang-eup, Asan 31499, Republic of Korea; 2Haematology and Molecular Medicine, University of the Witwatersrand, Private Bag 3, Johannesburg 2050, South Africa; 3Water Research Group, Unit for Environmental Sciences and Management, North University, Private Bag X6001, Potchefstroom 2520, South Africa; 4Intertox, 600 Stewart St #1101, Seattle, WA 98101, USA; 5HCT Co., Ltd., Icheon 17383, Republic of Korea

**Keywords:** OECD test guideline 412 and 413, inhalation toxicity test, toxicokinetics, lung burden measurement, nanomaterials, reducing animal

## Abstract

**Simple Summary:**

Animal welfare considerations with the 3R principles, Replacement, Reduction, and Refinement (3Rs), have become more important in regulatory toxicity testing. Modifying OECD inhalation toxicity test guidelines 412 and 413 aim to minimize animal use and maximize the number of endpoints that are measured for bronchoalveolar lavage fluid and lung burden measurement in toxicokinetics studies.

**Abstract:**

The OECD test guidelines for animal experiments play an important role in evaluating the chemical hazards. Animal tests performed using OECD guidelines, especially when the good laboratory practice (GLP) principle is applied, reduce the duplication of toxicity testing and ensure the best mutual acceptance of data by the OECD’s Mutual Acceptance of Data (MAD). The OECD inhalation toxicity test guidelines 412 (28 days) and 413 (90 days) have been revised. These OECD guidelines now reflect the inclusion of nanomaterials and recent scientific and technological developments. In particular, these test guidelines aim to evaluate the bronchoalveolar lavage fluid in the lungs for objective toxicity evaluation, along with the existing subjective histopathological evaluation. For solid particles, the lung burden measurement of particles is required for toxicokinetic studies and, in order to properly perform a toxicokinetic study, two post-exposure observations are recommended. In light of the revised OECD guidelines, we propose a method to reduce the number of animals when testing is conducted for nanomaterials.

## 1. Introduction

The OECD Test Guidelines (TGs) for testing chemicals are a collection of the most relevant internationally agreed testing methods used by governments, industry, and independent laboratories to assess the safety of chemicals. The TGs detail the procedures for regulatory safety testing, chemical notification, and registration. As scientific progress and country regulations continue to evolve, so do the TGs. Another component of OECD TGs is the mutual acceptance of data (MAD), which helps to address governmental requirements and provides a basis for cooperation among national authorities [1].

One of the critical principles of the experimental testing of chemicals is called Good Laboratory Practice (GLP), and ensures experimental data quality and integrity [2]. Those abiding by the principles of GLP in one OECD country must be accepted by other OECD countries for assessment purposes. The concept of “tested once, accepted for assessment everywhere” saves the chemicals industry the expense of duplicate testing for products marketed in more than one country [2].

The OECD revised the test guidelines for the existing inhalation toxicity tests to reflect the emergence of nanomaterials and recent scientific trends. In particular, the objective pathological evaluation was made mandatory by the addition of bronchoalveolar lavage fluid (BALF) analysis, breaking away from the dependence on subjective histopathology in the existing inhalation toxicity test evaluation. In addition, the revised test guidelines required measurements of the test article to be taken in real-time, along with off-line aerosol measurements in an inhalation chamber, reflecting the latest technology for the characterization of nanomaterials. Furthermore, the revised test guidelines (TG 412 and 413) require lung burden measurements to be conducted for solid particles during post-exposure observation periods to evaluate the lung clearance kinetics of the solid test particles [3,4].

In regulatory testing, the principle of Replacement, Reduction, and Refinement (3Rs) with REACH Article 4 of Directive 2010/63/EU states that “The Member States shall ensure that the number of animals used in projects is reduced to a minimum without compromising the objectives of the project”, and Article 13 states that “In choosing between procedures, those which to the greatest extent meet the requirements of the use of the minimum number of animals” [5]. We summarize the recent findings on lung burden studies using soluble and less soluble nanoparticles as a means to reduce animal use for OECD inhalation studies.

## 2. Minimizing Animal Use and Maximizing Endpoint Measured in OECD Test Guideline on Inhalation Studies

The current OECD TGs for inhalation toxicity testing 412 (28-day) and 413 (90-day) were recently revised to evaluate lung inflammation with objective measurements as well as to assess the lung burden of insoluble nanomaterials. Subsequently, additional animals are required for use immediately after termination of exposure for post-exposure observation periods to conduct bronchoalveolar lavage (BAL) fluid analysis and lung burden measurements. Measuring these variables, however, increases the number of testing subjects to 120 for TG 412 and 160 for TG 413, which is a large increase in testing subjects when compared to earlier testing guidelines (e.g., 40 for TG 412 and 80 animals for TG 413; see Table 1 and Table 2). Specifically, the guidelines recommend the use of a left lung for histopathology, a right lung for BAL fluid analysis, and a whole lung for lung burden measurements.

To comply with this directive, the revised TGs require additional animals. Increasing the number of animals increases the animal use and financial burden for all stakeholders (e.g., food, water, waste, disposal, renovation, or introduction of an inhalation chamber to accommodate the increased animal number). Regarding the test article exposure system, whole-body-type exposure chambers may require an increase in chamber volume to meet the requirement that “the total volume of the test animals should not exceed 5% of the chamber volume.” Nose-only-type chambers need more exposure ports, which can lead to an investment of capital by researchers. OECD recognizes this burden and seeks high quality and cost-effective research [6].

Recent inhalation studies by our group suggest that animal use in the test guideline studies can be minimized while maximizing the number of measured endpoints. A closer look at rat lung anatomy provides a useful opportunity to consider fewer animals that follow TG 412 and TG 413. The rat lung has five lobes: four lobes in the right lung (RCr, right cranial; RM, right median; RCa, right caudal; RA, right accessory) and one lobe in the left lung (Figure 1). We propose that the right lung(s) be used for histopathology and BAL measurements, and the one left lung lobe may be used for lung burden measurements. 

Nanoparticle testing provides evidence for this proposed approach. Nano-objects provide a test material that can be confidently measured in the lungs. In the subacute inhalation study with soluble silver nanoparticles (AgNPs) and insoluble gold nanoparticles (AuNPs), Park et al. [7] and Kim et al. [8] report an even lobar deposition of these nanoparticles in the lung lobes, measured as particle mass per gram of lung tissue. Soluble AgNPs or insoluble AuNPs were evenly deposited in the different lung lobes of both the right and left lungs (Table 3). These two studies provide evidence that any lung lobe can be used for lung burden measurement. We conclude that any of the rat lung lobes can be used for lung burden analysis to determine the deposited or retained total lung burden after short-term inhalation, certainly for these NPs, likely for nanoparticles of similar chemical/physical characteristics, and possibly with additional conventional chemicals. Taking this dataset further, other lung lobes can be used to collect and analyze BALF and histopathology. By using the rat lung lobes in our proposed manner, we spare 40 animals in TG 412 and 413 tests up to post-exposure observation (PEO)-2, and this offers the chance to conduct an additional BAL fluid analysis at PEO-3 (Table 4 and Table 5).

As we expressed above, the even lobar deposition of fibrous nanomaterials such as carbon nanotubes and plate-shape nanomaterials such as graphenes has not been studied to date. However, recent work by Devoy et al. [9] and Kim et al. [10] used one lobe for the lung burden measurement of carbon nanotubes and tangled multi-walled carbon nanotubes to assess clearance kinetics from the lung. Although the evenness of lobar deposition was not evaluated in these studies, the results suggest that the lung clearance kinetics could be evaluated by consistently using one lobe of the right lung. Current improvements in analyzing nanomaterial tissue (e.g., AgNP, AuNP, and CNTs) concentration are sensitive enough not to use a great quantity of the tissue, so one lobe of the right lung would suffice to analyze the amount of the retained nanomaterials.

**Figure 1 animals-12-03305-f001:**
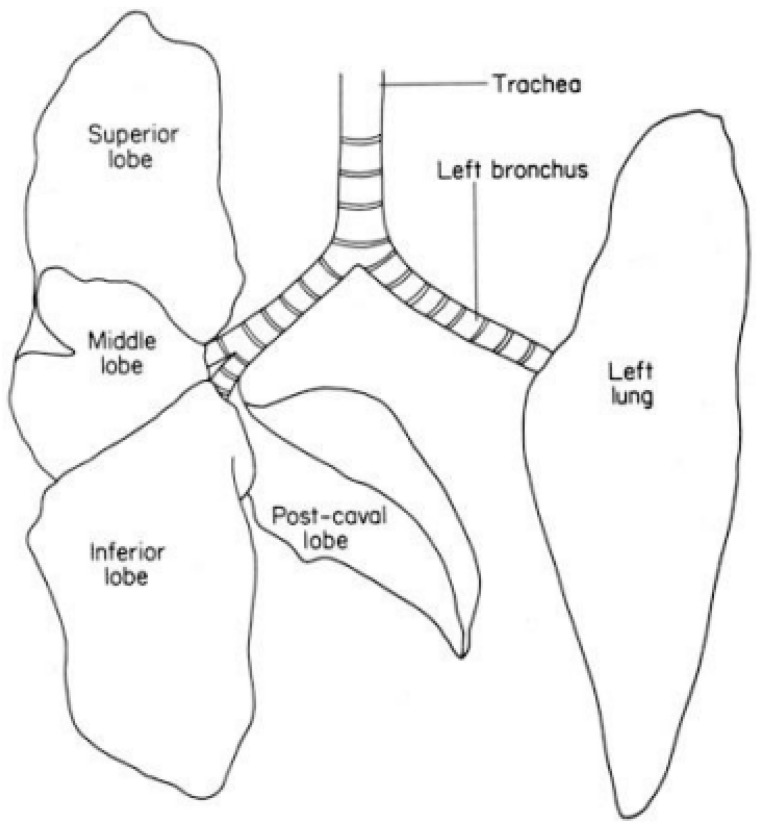
Rodent trachea and lungs (adapted from Figure 2 of OECD GD 39 [11]). Superior lobe (RC, right cranial); middle lobe (RM, right median); inferior lobe (RCa, right caudal), and post-caval lobe (RA, right accessory).

## 3. Reducing the Number of Animals When Conducting the Toxicokinetic Study for Nanomaterials after Inhalation Exposure

We believe that careful consideration of the lung tissues of each animal will lead to a reduction in the use of animals and maintain experimental integrity. To comply with TG 412 and 413, lung burden measurement is needed to derive clearance kinetics, and tissue distribution. Organs other than the lungs are used for both histopathological assessment and analytical chemical evaluation, assuming proper study design. We believe that toxicokinetic assessment, at least for nanomaterial inhalation exposure, can be conducted within the framework of current TG 412 and 413 with three sampling timepoints. 

Several considerations regarding the revision of toxicokinetic test guidelines are considered and include: (1) the number of samples for analysis and (2) analytical timepoints considering animal welfare and kinetic derivation. Based on our laboratory experiences and the literature, obtaining tissue distribution and clearance kinetic parameters from lungs and other tissues indicate that at least three sampling times, including PEO-1 (immediately after exposure), PEO-2- and PEO-3, and 4–5 animals per sampling point is recommended [8,12,13,14]. 

Most insoluble nanomaterials are eliminated from the lung by first-order elimination kinetics [12,14,15]. Three sampling timepoints, as prescribed by TGs 412 and 413, allow for the used data to estimate elimination half-time (T_1/2_) and elimination rate. Soluble nanomaterials such as AgNP have been known to follow a two-compartment model of fast and slow dissolution rates [13,14]. For soluble nanoparticles, three sampling points may not be inappropriate for the estimation toxicokinetic parameters due to the changing inflection point rising from different sampling times. Despite there being insufficient sampling points to estimate toxicokinetic parameters for soluble nanomaterials with a two-compartment model, toxicokinetic parameters can still be estimated. In doing so, inhalation toxicology studies for toxicodynamic and toxicokinetic can be performed with the 3Rs (replace, reduce, and refine) principle.

We understand that the current toxicokinetic TG 417 explicitly states that the test guideline is not intended for the testing of nanomaterials [16], the TG 417 is under collaborative revision efforts of OECD WPMN and EU NanoHarmony projects. Due to OECD WPMN and UE NanoHarmony, TG 417 will, at some point, include the evaluation of nanomaterials [17]. Insoluble nanomaterials are a priority consideration. 

## 4. Conclusions

If we follow the revised OECD inhalation toxicity test guidelines, more test animals are required. Specifically, TG 412 and 413 require lung clearance of solid particles and the BALF assay to objectively evaluate lung toxicity. Following the guidelines, 60–80 more animals are needed, as we demonstrate. Based on our laboratory’s work, we demonstrate that nanomaterials are uniformly deposited in multiple lobes in the lung burden. The remaining right lung lobes can be used for BALF analysis. The left lung can be used for histopathological evaluation. We see this as an opportunity to reduce animal use for OECD inhalation studies.

## Figures and Tables

**Table 1 animals-12-03305-t001:** Study design for TG 412.

	Revised OECD TG 412			Previous OECD TG 412
Exposure Group	Main Study (PEO-1)	PEO-2	PEO-3	Main Study
Control	HP (LL) + BAL (RL) = 5 M + 5 F LB (RL) = 5 M	HP (LL) + BAL (RL) = 5 F HP (LL) + LB (RL) = 5 M	LB (RL) = 5 M	HP (LL + RL) = 5 M + 5 F
Low	HP (LL) + BAL (RL) = 5 M + 5 F LB (RL) = 5 M	HP (LL) + BAL (RL) = 5 F HP (LL) + LB (RL) = 5 M	LB (RL) = 5 M	HP (LL + RL) = 5 M + 5 F
Moderate	HP (LL) + BAL (RL) = 5 M + 5 F LB (RL) = 5 M	HP (LL) + BAL (RL) = 5 F HP (LL) + LB (RL) = 5 M	LB (RL) = 5 M	HP (LL + RL) = 5 M + 5 F
High	HP (LL) + BAL (RL) = 5 M + 5 F LB (RL) = 5 M	HP (LL) + BAL (RL) = 5 F HP (LL) + LB (RL) = 5 M	LB (RL) = 5 M	HP (LL + RL) = 5 M + 5 F
	40 M + 20 F = 60	20 F + 20 M = 40	20 M	20 M + 20 F
Total	80 M + 40 F = 120			20 M + 20 F = 40

PEO, post-exposure observation; HP, histopathology; BAL, bronchoalveolar lavage; LL, left lung; RL; right lung; LB, lung burden; M, male; F, female.

**Table 2 animals-12-03305-t002:** Study design for TG 413.

	Revised OECD TG 413			Previous OECD TG 413
Exposure Group	Main Study (PEO-1)	PEO-2	PEO-3	Main Study
Control	HP (LL) + BAL (RL) = 10 M + 10 F LB (RL) = 5 M	HP (LL) + BAL (RL) = 5 F HP (LL) + LB (RL) = 5 M	LB (RL) = 5 M	HP (LL + RL) = 10 M + 10 F
Low	HP (LL) + BAL (RL) = 10 M + 10 F LB (RL) = 5 M	HP (LL) + BAL (RL) = 5 F HP (LL) + LB (RL) = 5 M	LB (RL) = 5 M	HP (LL + RL) = 10 M + 10 F
Moderate	HP (LL) + BAL (RL) = 10 M + 10 F LB (RL) = 5 M	HP (LL) + BAL (RL) = 5 F HP (LL) + LB (RL) = 5 M	LB (RL) = 5 M	HP (LL + RL) = 10 M + 10 F
High	HP (LL) + BAL (RL) = 10 M + 10 F LB (RL) = 5 M	HP (LL) + BAL (RL) = 5 F HP (LL) + LB (RL) = 5 M	LB (RL) = 5 M	HP (LL + RL) = 10 M + 10 F
	60 M + 40 F = 100	20 M + 20 F = 40	20 M	40 M + 400 F
Total	100 M + 60 F = 160			40 M + 40 F = 80

PEO, post-exposure observation; HP, histopathology; BAL, bronchoalveolar lavage; LL, left lung; RL; right lung; LB, lung burden; M, male; F, female.

**Table 3 animals-12-03305-t003:** Lung deposition of 5-day subacute inhalation exposure for soluble silver nanoparticles (AgNP, 20 nm) and insoluble gold nanoparticles (AuNP, 11 nm). Park et al. [7] and Kim et al. [8].

	AgNP		AuNP	
Lobes	Retention (μg/g of Lung Tissue)	Retention (μg/lobe)	Retention (μg/g of Lung Tissue)	Retention (μg/lobe)
RCr	28.41 ± 3.88	4.22 ± 0.84 (11.27%)	11.13 ± 1.44	1.23 ± 0.21 (7.85%)
RM	36.21 ± 2.68	5.58 ± 0.60 (14.88%)	15.64 ± 2.45	2.17 ± 0.38 (13.86%)
RCa	31.18 ± 3.14	10.89 ± 0.82 (29.06%)	14.85 ± 1.98	4.70 ± 0.49 (30.06%)
RA	36.82 ± 5.01	4.89 ± 0.50 (13.4%)	14.70 ± 3.93	1.84 ± 0.46 (11.77%)
RL	32.59 ± 2.01	25.58 ± 1.43 (68.26%)	14.39 ± 2.12	9.93 ± 0.96 (63.54%)
LL	26.80 ± 7.6	11.90 ± 3.02 (31.74%)	13.76 ± 2.63	5.70 ± 0.55 (36.46%)
Total lung	30.49 ± 3.81	14.12 ± 2.03	14.12 ± 2.03	15.63 ± 1.33

RCr; right cranial, RM; right median, RCa; right caudal, RA; right accessory lobe, LL; left lung; RL, right lobe; LL, left lobe.

**Table 4 animals-12-03305-t004:** Study design for TG 412 considering animal welfare.

	Revised OECD TG 412		
Exposure Group	Main Study (PEO-1)	PEO-2	PEO-3
Control	HP (LL) + [BAL (RL) +LB 1 lobe] = 5 M + 5 F	HP (LL) + [BAL (RL) +LB 1 lobe] = 5 M	HP (LL) + [BAL (RL) + LB 1 lobe] = 5 M
Low	HP (LL) + [BAL (RL) + LB 1 lobe] = 5 M + 5 F	HP (LL) + [BAL (RL) + LB 1 lobe] = 5 M	HP (LL) + [BAL (RL) + LB 1 lobe] = 5 M
Moderate	HP (LL) + [BAL (RL) + LB 1 lobe] = 5 M + 5 F	HP (LL) + [BAL (RL) + LB 1 lobe] = 5 M	HP (LL) + [BAL (RL) + LB 1 lobe] = 5 M
High	HP (LL) + [BAL (RL) + LB 1 lobe] = 5 M + 5 F	HP (LL) + [BAL (RL) + LB 1 lobe] = 5 M	HP (LL) + [BAL (RL) + LB 1 lobe] = 5 M
	20 M + 20 F0	20 M = 20	20 M
Total	60 M + 20 F = 80		

PEO, post-exposure observation; HP, histopathology; BAL, bronchoalveolar lavage; LL, left lung; RL; right lung; LB, lung burden; M, male; F, female.

**Table 5 animals-12-03305-t005:** Study design for TG 413 considering animal welfare.

	Revised OECD TG 413		
Exposure Group	Main Study (PEO-1)	PEO-2	PEO-3
Control	HP (LL) + [BAL (RL) + LB 1 lobe] = 10 M + 10 F	HP (LL) + [BAL (RL) + LB 1 lobe] = 5 M	HP (LL) + [BAL (RL) + LB 1 lobe] = 5 M
Low	HP (LL) + [BAL (RL) + LB 1 lobe] = 10 M + 10 F	HP (LL) + [BAL (RL) + LB 1 lobe] = 5 M	HP (LL) + [BAL (RL) + LB 1 lobe] = 5 M
Moderate	HP (LL) + [BAL (RL) + LB 1 lobe] = 10 M + 10 F	HP (LL) + [BAL (RL) + LB 1 lobe] = 5 M	HP (LL) + [BAL (RL) + LB 1 lobe] = 5 M
High	HP (LL) + [BAL (RL) + LB 1 lobe] = 10 M + 10 F	HP (LL) + [BAL (RL) + LB 1 lobe] = 5 M	HP (LL) + [BAL (RL) + LB 1 lobe] = 5 M
	40 M + 40 F = 100	20 M	20 M
Total	80 M + 40 F = 120		

PEO, post-exposure observation; HP, histopathology; BAL, bronchoalveolar lavage; LL, left lung; RL; right lung; LB, lung burden; M, male; F, female.

## Data Availability

Not applicable.

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
