# Peer review of "Animal Welfare Considerations When Conducting OECD Test Guideline Inhalation and Toxicokinetic Studies for Nanomaterials"

_animals, 2022, doi:10.3390/ani12233305_

Round 1
Reviewer 1 Report
-
The article has novel ideas and precise entry points,
-
but the conclusion is a little too elliptical,
-
which should to continue to improve.
Author Response
- No comments: Nothing changed, but we changed several texts to reflect other reviewers' comments.
Reviewer 2 Report
The manuscript is well written and article sheds information in order to limit the use of animals in OECD inhalation experiments, briefly explained the most recent data on lung burden studies employing soluble and less soluble nanoparticles.
Line 28 - Correct this ".."
Line 45 - Please elaborate "GLP"
Line 144 - Section-3 is not clear. The heading says "saving animals" used in the heading and it was not clearly explained in the text. Please include a statement clearly mentioning why "Saving animals mentioned", it will be good to follow the main objective of this manuscript.
Author Response
- The manuscript is well written and article sheds information in order to limit the use of animals in OECD inhalation experiments, briefly explained the most recent data on lung burden studies employing soluble and less soluble nanoparticles. Nothing changed
- Line 28 - Correct this ".." Corrected.
- Line 45 - Please elaborate "GLP" Updated text to provide support for GLP. Also added reference. Also noted incorrect reference for cost related text. Changed OECD 2022 to OECD 2019 to be consistent with reference list.
- The manuscript is well written and article sheds information in order to limit the use of animals in OECD inhalation experiments, and briefly, explained the most recent data on lung burden studies employing soluble and less soluble nanoparticles.
- Response:Nothing changed
- Line 28 - Correct this ".." Corrected.
- Line 45 - Please elaborate "GLP"
- Response: Updated text to provide support for GLP. Also added reference. Also noted incorrect reference for cost related text. Changed OECD 2022 to OECD 2019 to be consistent with reference list.
- Line 144 - Section-3 is not clear. The heading says "saving animals" used in the heading and it was not clearly explained in the text. Please include a statement clearly mentioning why "Saving animals mentioned", it will be good to follow the main objective of this manuscript.
- Response: Updated text to provide support for the concept of “saving animals.” First, we will not use “saving animals” and so the title of Section 3 reads: “Reducing the number of animals when conducting the toxicokinetic study for nano-materials after inhalation exposure.” Second, we changed the order of paragraph to better illustrate our point.
Reviewer 3 Report
The paper proposed alternative methodology for animal studies, in particular evaluating atmospheric toxicants by inhalation. The paper is well described and discussed by using the authors" experiment data.
Major points
Generally, BAL procedure use saline. We usually examine BAL at first, and send the lavaged lungs to histology by hormaldehyde fixation or homogenate. Please reconsider the method.
Minor points
P3, L99; soluble silver nanoparticles (AgNPs)......
Author Response
- The paper proposed alternative methodology for animal studies, in particular evaluating atmospheric toxicants by inhalation. The paper is well described and discussed by using the authors" experiment data.
- Response: Nothing changed
- Generally, BAL procedure use saline. We usually examine BAL at first, and send the lavaged lungs to histology by formaldehyde fixation or homogenate. Please reconsider the method.
- Response: You are correct when you are doing BAL for general lung toxicity evaluation for gas, vapor, or soluble particles. However, when you are doing the lung burden measurement of solid particles along with BAL, the BAL procedure may wash out the solid particles deposited in the lung airways. The guideline recommends a separate procedure for BAL and lung burden measurement to avoid underestimated lung burden measurement.
- P3, L99; soluble silver nanoparticles (AgNPs)......
- Response: corrected